# Cysteine Oxidation Promotes Dimerization/Oligomerization of Circadian Protein Period 2

**DOI:** 10.3390/biom12070892

**Published:** 2022-06-25

**Authors:** Fernando Martin Baidanoff, Laura Lucía Trebucq, Santiago Andrés Plano, Phillip Eaton, Diego Andrés Golombek, Juan José Chiesa

**Affiliations:** 1Laboratorio de Cronobiología, Departamento de Ciencia y Tecnología, Universidad Nacional de Quilmes/CONICET, Bernal B1876BXD, Argentina; fbaidanoff@gmail.com (F.M.B.); laura.trebucq@gmail.com (L.L.T.); dgolombek@gmail.com (D.A.G.); 2Institute for Biomedical Research (BIOMED), Catholic University of Argentina/CONICET, Buenos Aires C1107CABA, Argentina; splano@gmail.com; 3William Harvey Research Institute, Queen Mary University of London, London EC1M 6BQ, UK; p.eaton@qmul.ac.uk

**Keywords:** redox, circadian clock, S-nitrosation, PER2

## Abstract

The molecular circadian clock is based on a transcriptional/translational feedback loop in which the stability and half-life of circadian proteins is of importance. Cysteine residues of proteins are subject to several redox reactions leading to S-thiolation and disulfide bond formation, altering protein stability and function. In this work, the ability of the circadian protein period 2 (PER2) to undergo oxidation of cysteine thiols was investigated in HEK-293T cells. PER2 includes accessible cysteines susceptible to oxidation by nitroso cysteine (CysNO), altering its stability by decreasing its monomer form and subsequently increasing PER2 homodimers and multimers. These changes were reversed by treatment with 2-mercaptoethanol and partially mimicked by hydrogen peroxide. These results suggest that cysteine oxidation can prompt PER2 homodimer and multimer formation in vitro, likely by S-nitrosation and disulphide bond formation. These kinds of post-translational modifications of PER2 could be part of the redox regulation of the molecular circadian clock.

## 1. Introduction

In mammals, circadian (i.e., circa 24-h) rhythms in most cellular and physiological processes are regulated by a network of circadian oscillators. The cellular circadian clockwork is composed of molecular self-sustained oscillations of several transcriptional/translational feedback loops. Circadian locomotor output cycles kaput (CLOCK) and Brain and muscle Arnt-like 1 (BMAL1, or ARNTL-1) are basic helix–loop–helix (bHLH) transcription factors which contain Per-Arnt-Sim (PAS) domains and can heterodimerize to activate period (per) and cryptochrome (cry) genes [1]. In turn, PER1-2 and CRY1-2 proteins accumulate in the cytoplasm to form heterodimers that inhibit the activity of CLOCK and BMAL1 to re-start the cycle. The CLOCK:BMAL1 heterodimer also activates the orphan nuclear receptor rev-erb α/β (REV-ERB α/β) and ror α/β genes, generating an additional feedback loop which stabilizes the oscillation.

The kinetics of PERs:CRYs and CLOCK:BMAL1 heterodimers and the stoichiometry of its monomer components are key factors setting the molecular timing of the circadian clock. For example, PER: PER dimers have been described as regulating behavior and molecular rhythmicity in flies [2], and a similar interaction between mouse PER proteins regulates their stability and cellular localization in vitro and in vivo [3,4]. Post-translational control of the stability of PERs includes phosphorylation by casein kinases 1 (CK1) δ and ε, leading to protein accumulation by increasing stability, or to degradation by ubiquitination pathway, depending on the specific phosphosites [5]. Of importance for this study, the redox balance can regulate the activity and stability of clock proteins through different mechanisms (e.g., heme binding on REV-ERB α/β [6]; zinc binding and disulfide bound formation stabilizing PER2:CRY1 heterodimer [7]). Indeed, the molecular oscillator is affected by hydrogen peroxide cysteine thiol oxidation of CLOCK, prompting its interaction with BMAL1 [8]. Cysteine oxidation may also occur through an S-nitrosation reaction, generating an S-nitrosothiol. This post-translational modification was shown to change the quaternary structure of proteins, affecting enzymatic activity and several pathways, and was hypothesized to participate in the photic circadian signaling [9].

In this context, we tested whether the clock protein PER2 in HEK-293T cells can be oxidized by S-nitrosation, thereby leading to changes in the stability of its monomer. HEK-293T cells were selected due to their extensive use as an established model for studying the circadian molecular clock regulation (e.g., resetting with dexamethasone [10], peroxide control of CLOCK cysteine oxidation on CLOCK-BMAL1 heterodimer activity [8], or its outputs (e.g., proteasome activity on oxidized proteins [11], enzymatic function in oxidative stress [12]). Our results show that PER2 exhibits cysteines susceptible to oxidation and that treatment with the S-nitrosating compound S-nitroso-L-cysteine (CysNO), or with hydrogen peroxide, reduced its monomer, subsequently increasing homodimers and oligomers in a time- and dose-dependent manner. These changes in the stability of PER2 protein induced by reversible cysteine oxidation could be of importance for regulating the circadian oscillator activity.

## 2. Materials and Methods

### 2.1. Synthesis of S-nitroso-L-cysteine

S-nitroso-L-cysteine (CysNO) was synthesized just before each assay. Briefly, equimolar amounts (100 mM) of L-cysteine (Fluka Analytical, Buchs, Switzerland) in 0.1 N HCl and (100 mM) sodium nitrite (SIGMA, St. Louis, MO, USA) were reacted under darkness for 10 min, adding 0.1 N NaOH to stop the reaction. Generation of CysNO was determined by spectrophotometer at 340 nM [13].

### 2.2. Cell Cultures and Transfections

Human embryonic kidney 293T (HEK-293T) line cells were cultured in Dulbecco’s Modified Eagle Medium (DMEM Gibco^®^, ThermoFisher^®,^ Waltham, MA, USA) high glucose anti-anti (Gibco^®,^ ThermoFisher^®,^ Waltham, MA, USA) in 25 cm^2^ flasks kept in a humidified incubator at 37 °C, 95% air/5% CO_2_ up to confluence. Sub-cultures were obtained every 4–5 days, at a rate of 1:3–1:6 by washing with phosphate buffer saline (PBS, pH 7.4), incubating with 0.05% Try psin/EDTA (Gibco^®^) for 10 min at 37 °C, taking 2 × 10^6^ cells/flask. About 10^5^ cells/well were cultured into 12-well dishes for further transfection and pharmacological treatment. pCMV Sport2 mPer2 plasmid was generously provided by Cheng Lee (Addgene, ThermoFisher^®^, Waltham, MA, USA, #16204 [14]).

### 2.3. Polyethylene Glycol (PEG)-Switch Assay

Protein thiol modifications of target cysteines susceptible to reversible oxidation were determined by a PEG-based alkylating agent altering electrophoretic mobility of proteins [15]. Transfected HEK-293T cells were treated with hydrogen peroxide (H_2_O_2_, 10, 100 or 1000 µM) for 10 min. Then, they were lysed with 100 mM maleimide 1% SDS in 100 mM Tris-HCl (pH 7.4) buffer and incubated for 25 min at 50 °C, blocking cysteines reduced to thiols. Next, 1–4 Dithiothreitol (DTT, Sigma^®^) was added (final concentration of 200 mM) for 30 min allowing the reduction of cysteines. Then, reduced cysteines were reacted with Metoxi-PEG-maleimide 5000 or 10,000 (Sigma^®^), achieving 10 mM concentration in 1% SDS for 2 h. Finally, the reaction was stopped with 4X Laemmli buffer with 10% 2-mercaptoethanol (Sigma^®^), and samples were loaded into SDS-PAGE for electrophoretic mobility assay and processed for Western blot. PEG-tagged proteins will increase molecular weight, thereby diminishing mobility with respect to control.

### 2.4. Biotin-Switch Assay

Oxidative S-nitrosations were detected in protein homogenates through the selective reduction of S-nitrosothiols with ascorbate and its further biotinylation, i.e., a biotin switch assay [16]. Briefly, HEK-293T cells overexpressing PER2 were treated with 100 µM CysNO or PBS for 1, 3, 5, 10, 15 and 30 min. Then, they were washed-out with PBS, lysed (1% SDS, 100 mM maleimide, 0.2 mM neocupine, 1 mM diethylene-triamine-penta-acetic acid, 100 mM Tris-HCl, pH 7.4) and incubated for 25 min at 50 °C under darkness to block free thiols. Samples were then desalted in Zeba Spin columns (Thermo Scientific) and eluates were collected in the tagging buffer (0.5 mM biotin-maleimide, 0.5% SDS, 10 µM copper sulphate (II), 60 mM ascorbate) and incubated for 1 h under darkness at room temperature. To stop the reaction, 100 mM maleimide in 100 mM Tris-HCl, pH 7.4 was used. Taking the molecular weight for PER2 (150 kDa), samples were concentrated by centrifugation and filtration at 100 kDa (Amicon®, Merck- Millipore, Darmstadt, Germany). Finally, biotinylated proteins were precipitated with magnetic beads conjugated with streptavidin (Dynabeads^®^ MyOne™ Streptavidin T1, Invitrogen^®,^ ThermoFisher^®,^ Waltham, MA, USA) and dissolved in 4× Laemmli buffer for further analysis by Western blot.

### 2.5. Characterization of the Cysteine Oxidation Reaction

Cysteine oxidation by the CysNO, altering the PER2 monomer, was tested for reversibility. HEK-293T cells were treated with 100 μM CysNO or PBS for 30 min, and then lysed with 1% SDS in 100 mM Tris-HCl (pH 7.4) buffer. Homogenates were then exposed to 10% 2-mercaptoethanol (or buffer alone), supplying a reductive environment for testing the reversibility of cysteine oxidation on PER2 monomer in Western blots. Since CysNO can decompose to NO and L-cysteine, the effects of L-cysteine on the PER2 monomer were also controlled by treating HEK-293T cells with 100 μM L-cysteine, or 100 μM CysNO, for 30 min, and then they were lysed, and homogenates were analyzed in Western blots for PER2.

### 2.6. Western Blot

Samples for Western blot were previously diluted in corresponding loading buffer. The following elements were used: pre-cast Gels (Mini-PROTEAN^®^ TGX™, BioRad^®,^ Hercules, CA, USA) 4–20% or 4–15%; molecular weight ladder (Precision Plus Protein Western Blotting Standards, BioRad^®^); Trans-Blot^®^ TurboTM Transfer System (BioRad^®,^ Hercules, CA, USA); Immun-Blot^®^ PVDF membranes (BioRad^®^, Hercules, CA, USA); pre-mounted Trans-Blot turbo Transfer (BioRad^®^, Hercules, CA, USA) (only for biotin-switch processed samples). Primary antibodies were employed in 1/1000 dilutions: rabbit isotype anti-PER2 (AB5428P, Merck-Millipore, Darmstadt, Germany), mouse anti-ACTIN (A5441, Sigma-Aldrich, Burlington, MA, USA) and rabbit anti-GAPDH (AB#2118m, Cell Signalling Technologies, Danver, MA, USA) in 0.05% Tween-20, 5% non-fat dry milk in PBS. Secondary antibodies conjugated with horseradish peroxidase anti-mouse or rabitt IgG (Cell Signalling Technologies, Danver, MA, USA) were used in 1/5000 dilutions. Pierce™ ECL Western Blotting (ThermoFisher^®^, Waltham, MA, USA) was used for revealing, or SuperSignal™ West Femto Maximum Sensitivity Substrate (ThermoFisher^®^, Waltham, MA, USA) for better resolution of biotin-switch processed samples. Finally, radiographs were scanned to obtain images for quantification of Intensity Optic Density by means of GelProAnalyzer version 3.1 software (Media Cybernetics^®^, Rockville, MD, USA).

## 3. Results

### 3.1. Cysteine Oxidation in PER2 and CRY2

PER2 and CRY2 circadian proteins were assessed for their ability to be oxidized in cysteine thiols. First, PEG-switch assayed-protein extracts from HEK-293T cells were tested for whether accessible cysteines were susceptible to oxidation (Figure 1A). A band displacement was observed even for the lower hydrogen peroxide concentration (when using a 10 kDa PEG). Increasing gel resolution, as well as using a higher molecular weight PEG, did not generate a greater displacement for PER2 (Appendix A). CRY2 showed a clear band displacement after treatment with 200 (but not with 100) µM hydrogen peroxide (Appendix A). The next step was treating cells with S-nitrosocysteine (CysNO), which efficiently induces trans-S-nitrosation of cysteines. HEK-293T cells treated with 10, 100 or 200 µM CysNO or PBS for 30 min were assayed by biotin-switch and labelled with streptavidin-HRP in Western blots (Figure 1B). Labelled proteins were found to be increased compared to the control for 100 and 200 µM CysNO. This allowed us to set the conditions for the specific detection of circadian proteins. However, treatment with 100 µM CysNO (for 1–5 min) allowed the labelling and detection of the S-nitrosated version of PER2, being unstable for its detection after 5 min, with no signal for S-nitrosated PER2 at 15 min (Figure 1C).

### 3.2. Time and Dose Response of PER2 to CysNO

Time-dependent changes of PER2 were studied after treatment with 100 µM CysNO (Figure 2A). The band corresponding to its monomer (150 kDa) decreased after exposure to CysNO, being undetectable after 10 min. A 250 kDa band appeared transiently, concomitantly with the disappearance of the monomer (1–5 min), together with a higher molecular weight band that was present throughout the complete time range studied (1–30 min). The 250 kDa band can be identified as the dimeric PER2:PER2 form, whereas the high molecular weight bands correspond to its multimeric aggregate. We also assessed the effect of the CysNO concentration on the PER2 monomer (Figure 2B). In the 0–100 μM range, a concentration-dependent effect was clearly observed in the Western blots for the decrement of monomers and formation of multimers, while dimerization was variable.

### 3.3. Characterization of the Cysteine Oxidation Reaction

The reaction of cysteine thiol with CysNO, which induced changes in the PER2 monomer, was tested as a reversible oxide-reduction, by post-treatment with the reductive agent 2-mercaptoethanol. The generation of PER2 dimers and multimers by CysNO was reversed after treatment with 2-mercaptoethanol (Figure 3A), which significantly abolished dimers and multimers, increasing monomer levels. In addition, the trans-S-nitrosation by CysNO was compared with the effect of L-cysteine as a control (L-Cys) (Figure 3B). Treatment with L-Cys did not generate changes on PER2 monomer levels as compared to CysNO (although aggregates were obtained under all treatments). In addition, changes on PER2 were tested by treating cells with the physiologic oxidant hydrogen peroxide (Figure 3C). The monomer 150 kDa band decreased as the hydrogen peroxide concentration increased, without changes in dimers and multimers.

## 4. Discussion

In this work, we assessed the ability of the circadian clock protein PER2 to react with the S-nitrosating CysNO agent and to generate reversible oxidations, thus changing monomer stability. PER2 (Figure 1A) (and CRY2, Appendix A) exhibited accessible cysteines in the PEG switch assay that can respond to oxidative conditions. This assay was performed under different conditions that failed to generate substantial displacement (see Appendix A). This is because PER2 is a protein of 130 kDa, which is a large molecular weight for this kind of assay. Moreover, increasing PEG molecular weight to 20 kDa is not recommended. Thus, we decided to include PEG switch for 10 kDa (Figure 1A), showing enough band displacement to decide to study PER2 as a candidate for cysteine oxidation experiments. S-nitrosation was assayed in cell proteins by biotin-switch and detected in proportion to the CysNO concentration (Figure 1B). Following this, S-nitrosated PER2 was successfully identified by the same assay (Figure 1C). Treatment with CysNO altered PER2 monomer stability in both a time- (Figure 2A) and dose- (Figure 2B) dependent manner. The PER2 monomer was transiently detected (0–30 min) under control conditions and decreased immediately after 100 μM CysNO treatment, together with the appearance of homodimers and multimers. These sequential changes from monomer-dimer–multimer were independent of the CysNO dose at the time selected for the assessment (30 min). However, a dose-dependence was found for the multimer increment—and partially for the monomer reduction—in the range from 10–100 μM, while a 1000 μM dose generated a complex band pattern formed by the three structures.

Formation of protein dimers and aggregates by cysteine reaction with CysNO was tested for reversibility, since as other oxidations, these are broadly determined by the redox potential. Dimers and aggregates generated after treatment with CysNO were not observed when homogenates were exposed to the reductant 2-mercaptoetanol (Figure 3A). Thus, it can be inferred that the reductant conditions, which normally impede disulfide formation, reversed those induced by CysNO. This low molecular weight nitrosothiol is widely used as a typical NO• donor to generate S-nitrosations [17] since it spontaneously decomposes in aqueous solutions to release NO•. Then, this characteristic of the S-nitroso reactant was tested, decreasing the levels of the PER2 monomer as compared to the effect of L-cysteine (Figure 3B). Brief (5–30 min) exposure to CysNO generates protein S-nitrosothiols in several cell types using L-type amino acid transmembrane transport [18,19,20], which mediates the movement of NO• equivalents into cells. In addition, oxidative changes in PER2 were also observed with hydrogen peroxide (Figure 3C), a powerful cellular oxidant that can react with accessible cysteines through peroxiredoxin–thioredoxin systems [21]. Indeed, a circadian oscillation of hydrogen peroxide was recently demonstrated to determine a redox oscillation of CLOCK through Cys195 thiol oxidation [8].

S-nitrosated proteins are inherently labile, with a propensity to spontaneously react with thiols to generate disulfides. Indeed, it has been proposed that protein S-nitrosation is predominantly an intermediate oxidative state of thiols that typically leads to disulfide formation [22]. By conjugating the proteins by disulfide bonds, they stabilize homo- and heterodimers or multimers that could lead to changes in signaling by protein interactions [23]. As examples, protein S-nitrosation by CysNO leading to disulfide-dependent dimerization was found for the protein kinase A regulatory subunit (PKAR) 1 and PKG 1α [22]; in addition, S-nitrosation followed by phosphorylation of caveolin 1 can also destabilize its oligomeric structure and function, leading to monomer accumulation and degradation [24]. Although disulfide formation was not demonstrated in this work, it could be of importance if redox derivatives (e.g., ROS/RNS), endogenous S-nitrosothiols (GSNO) and/or buffering metabolites (GSH) are taken as circadian messengers that change the structure, half-life and degradation of core circadian clock proteins, a possibility that was hypothesized [9] and remains to be tested.

Phosphorylation altering the dynamics and stability of clock proteins establishes the kinetics (formation/degradation rates) of the inhibitory complexes of the molecular oscillator, of which PER and CRY proteins represent the delayed negative feedback on clock gene expression and circadian timekeeping [1]. In addition, circadian non-transcriptional oscillations of peroxiredoxin 1–2 and thioredoxin activity were found in red cells [25], and this conserved redox oscillator was proposed as part of the circadian timekeeping mechanism [26]. Here, we suggest a putative redox regulatory component by cysteine S-nitrosation, leading to homodimerization–multimerization, likely altering the stability and half-life of clock proteins. In line with this idea, a recent study demonstrated that hydrogen peroxide is an input that can regulate the phase of PER2:LUC fibroblast rhythms through PER2 degradation [27]. The redox input to circadian timekeeping mechanism was also tested at the behavioral rhythm level [28], as well as at the hypothalamic SCN neurons [29], where the oxidant agent diamide affected both redox homeostasis and electrical activity rhythms.

Although the functional significance of such redox regulation is still unknown, some hypotheses can be proposed. NO• has been studied as a short-lived intracellular and paracrine messenger for the photic transduction of the clock at the hypothalamic suprachiasmatic nucleus through cGMP signaling [30]. Increasing NO• levels by light-activation of nNOS could destabilize the PER2:CRY1 heterodimer by zinc tetra-thiolate oxidation [7], allowing further PER2 homodimerization, multimerization and degradation. In addition, the monomer/homodimer ratio could establish the degradation kinetics of clock proteins, eventually changing the phase of the circadian clock. Photic degradation of BMAL1 could be of importance when competing with CLOCK for the CLOCK: BMAL1 [31] and/or NPAS2:BMAL1 [32] transcriptional activator complexes. Reversible oxidation by hydrogen peroxide of critical cysteine thiols is essential for determining CLOCK:BMAL1 interaction and the molecular clock period [8]. Our current results suggest an additional role for NO• (Figure 4), in which photic activation of this second messenger may induce S-nitrosation (modelled here by CysNO), affecting PER2 clock protein stability through dimerization/oligomerization, and hence the molecular clock oscillation.

## Figures and Tables

**Figure 1 biomolecules-12-00892-f001:**
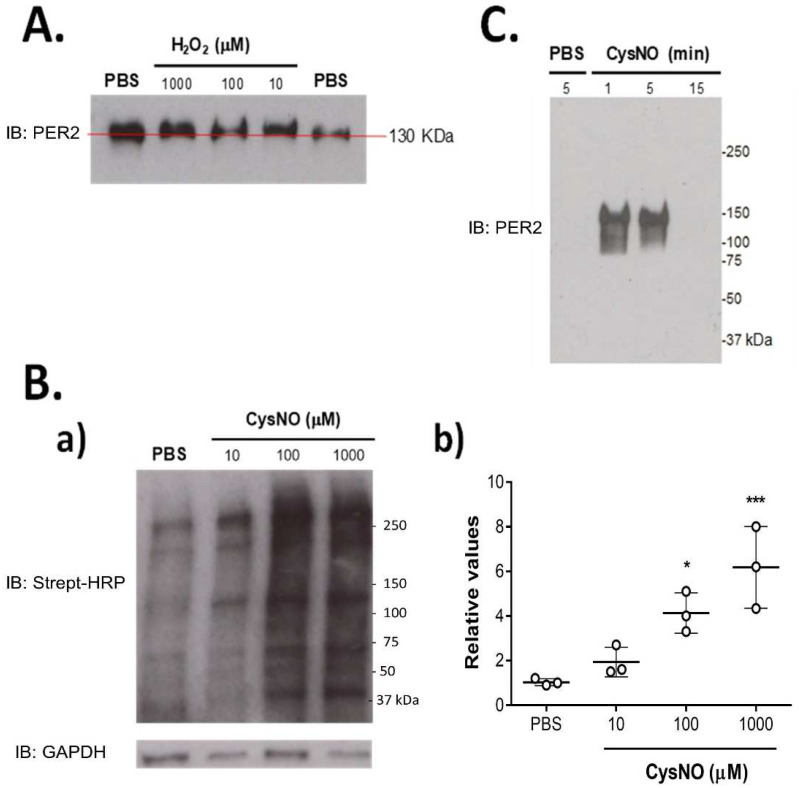
Assessment of cysteine oxidation of PER2 in HEK-293T cells. (**A**) Cells treated with PBS or 10, 100, 1000 µM hydrogen peroxide (H_2_O_2_) for 10 min and then assayed by PEG-switch. Band displacement corresponding to 10 kD PEG was observed for all concentrations. (**B**) Cells treated with PBS or 10, 100, 1000 µM CysNO for 30 min where assayed by biotin-switch. (**a**) Representative Western blot revealed for Streptavidin-HRP; (**b**) total IOD values relative to GAPDH for each lane (mean ± SEM, one-way ANOVA, *p* < 0.001, *t*-test * *p* < 0.05, *** *p* < 0.001 vs. PBS; *n* = 3). (**C**) Oxidized PER2 was detected with biotin-switch assay after 100 μM CysNO treatment (1–5 min).

**Figure 2 biomolecules-12-00892-f002:**
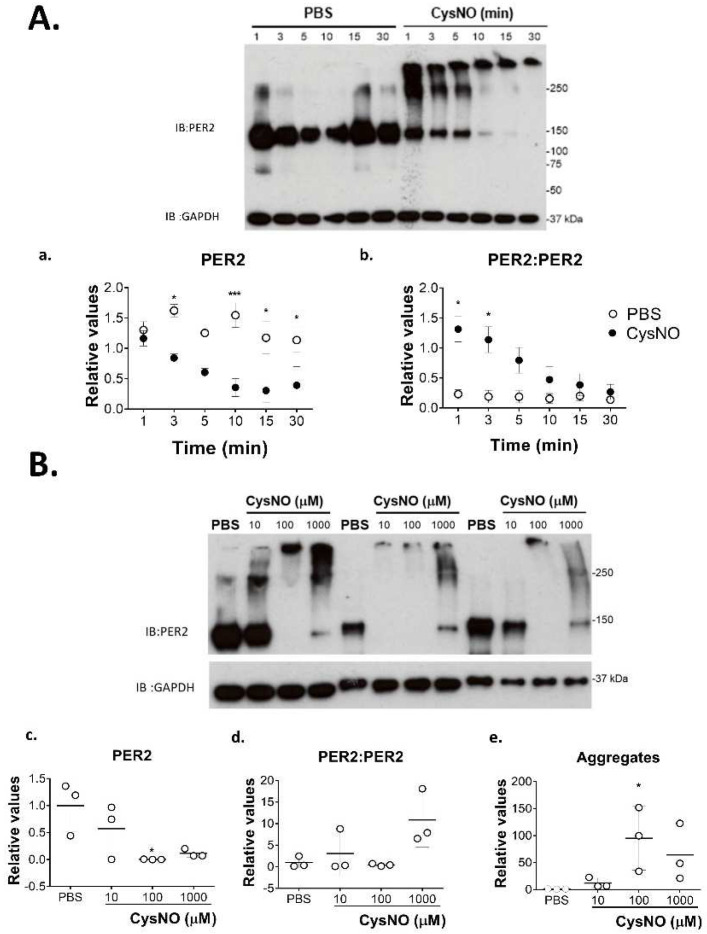
Effects of CysNO on PER2 dimerization/aggregation. (**A**) HEK-293T cells were treated with 100 μM CysNO or PBS for 1, 3, 5, 10, 15 or 30 min. Representative immunoblot for PER2 and GAPDH, showing bands corresponding to monomer PER2 (~150 kDa), homodimer (~250 kDa, PER2:PER2) and high molecular weight aggregates or multimers (>250 kD); GAPDH (~37 kDa). Relative values for PER2 (**a**) monomer and (**b**) homodimer, for each replicate of three experiments (values for aggregates are not shown due to high variability between individual blots). IOD values were relativized to the corresponding value for GAPDH (two-way ANOVA for: treatment, *p* < 0.0001; time *p* < 0.05; followed by Bonferroni post-hoc test, * *p* < 0.05 and *** *p* < 0.001; for PER2:PER2: two-way ANOVA for: treatment, *p* < 0.0001, time, *p* < 0.01, followed by Bonferroni post-hoc test, * *p* < 0.05). (**B**) Effects of CysNO concentration on PER2 dimerization/aggregation. Cells were treated with 10, 100, 1000 μM CysNO and PBS for 30 min. IOD for PER2 (~150 kD), PER2:PER2 (~250 kD) or aggregates were relativized to the corresponding IOD for GAPDH, and then to the average value for the control group (PBS) for each version. Relative values for (**c**) PER2, (**d**) PER2:PER2 or (**e**) aggregates for each replicate (for PER2 and aggregates: one-way ANOVA, *p* < 0.05, followed by Dunnet test, PBS vs. 100 μM, * *p* < 0.05, *n* = 3).

**Figure 3 biomolecules-12-00892-f003:**
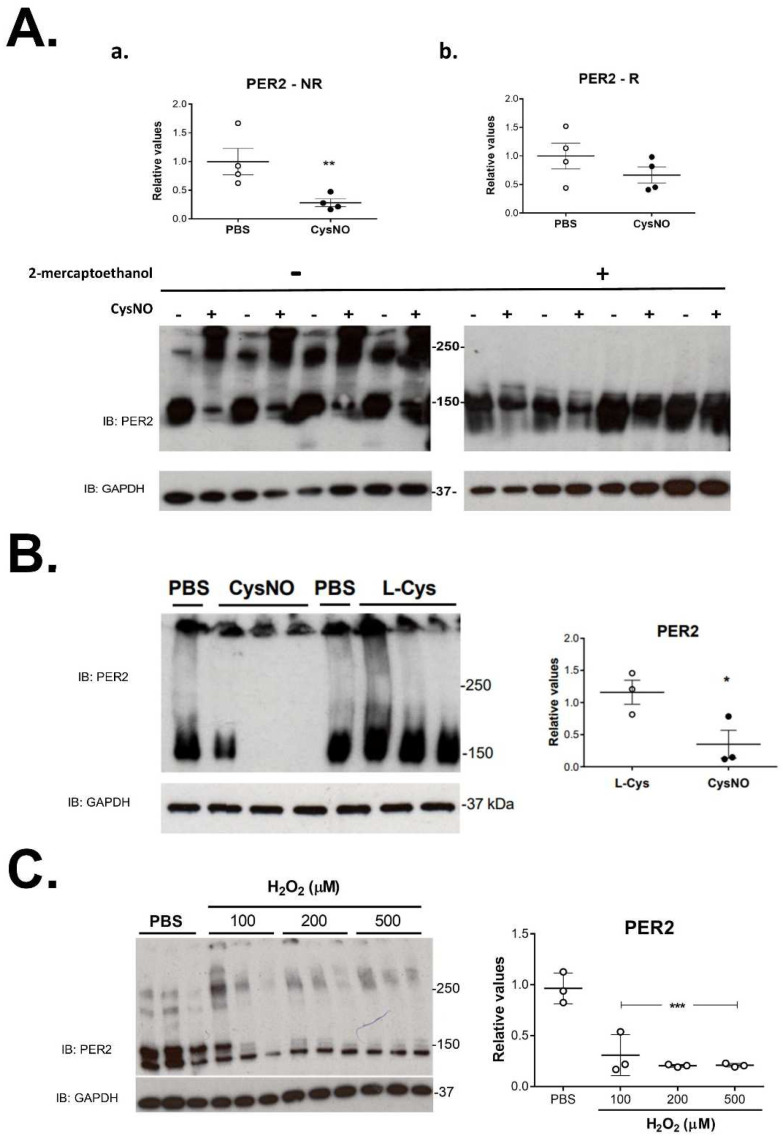
Characterization of PER2 oxidation reaction. Reversibility of CysNO treatment on PER2 dimerization/aggregation. (**A**) HEK-293T cells were treated with 100 μM CysNO or PBS for 30 min. Immunoblots (IB) for PER2 and GAPDH from samples treated without (non-reduced, NR) (left panel) or with 2-mercaptoethanol (reduced, R) (right panel). IOD for bands corresponding to PER2 (~150 kD), PER2:PER2 (~250 kD), or aggregates (>250 kD), were relativized to the corresponding value for GAPDH, and then to the average value for PBS. Values are mean ± SEM for: (**a**) PER2 for samples treated with 2-mercaptoethanol; (**b**) PER2 for control samples (*t*-test for independent samples, ** *p* < 0.01); (**B**) specificity of CysNO treatment on PER2 dimerization/aggregation. HEK-293T cells were treated with 100 μM CysNO or L-Cysteine (L-Cys) for 30 min. Immunoblot for PER2 showing bands corresponding to monomer and aggregates (left panel) IOD values for PER2 monomer relative to the GAPDH value for corresponding lane (mean ± SEM for PER2, *t*-test for independent samples, * *p* < 0.05) (right panel). (**C**) Treatment with hydrogen peroxide generated changes on PER2 dimerization/aggregation. HEK-293T cells were treated with 100, 200 or 500 μM hydrogen peroxide (H_2_O_2_) or PBS for 30 min. Immunoblot for PER2 showed bands corresponding to monomer, dimer, aggregates, and unspecific band (left panel); IOD values for PER2 monomer relative to GAPDH value for the corresponding lane (mean ± SEM for PER2 monomer, one-way ANOVA followed by Tukey’s post-hoc test, PBS vs. 100, 200 or 500 μM, *** *p* < 0.001) (right panel).

**Figure 4 biomolecules-12-00892-f004:**
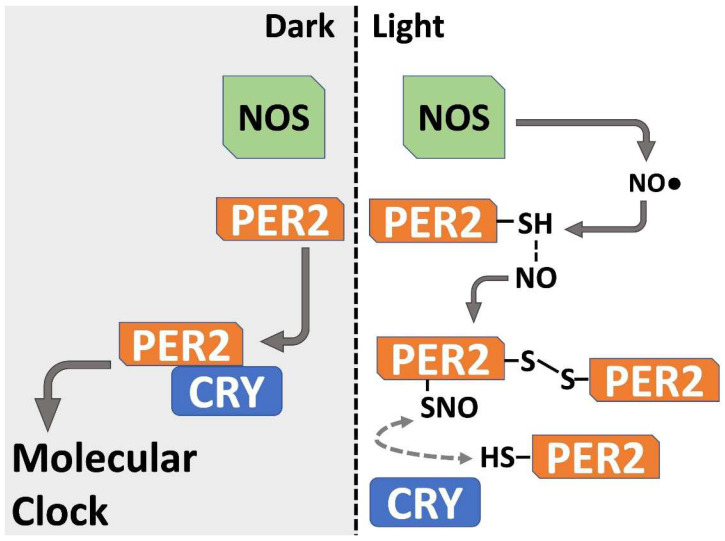
A framework in which cysteine oxidation of PER2 may participate in the control of the circadian molecular clock at the suprachiasmatic nucleus neurons. Light-induced Nitric Oxide Synthase (NOS) activation leads to an increase in nitric oxide (NO•) concentration. This molecule is able to interact as a second messenger through cGMP pathway, and also to oxidize accesible cysteines (-SH) in PER2 clock protein generating S-nitrosation (RSNO), intermolecular disulfide bond, dimerization/oligomerization, decreasing PER2 monomer stability, and thus the ability to form an heterodimer with CRY (the component of the “negative” loop of the core molecular circadian clock). As well as other physiological oxidants as hydrogen peroxide, this putative mechanism could lead to alteration of the molecular clock oscillation.

## Data Availability

All data of the article is available under request.

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
