# Peer review of "Cysteine Oxidation Promotes Dimerization/Oligomerization of Circadian Protein Period 2"

_biomolecules, 2022, doi:10.3390/biom12070892_

Round 1

Reviewer 1 Report

 In the present work, the ability of the circadian protein period 2 (PER2) to undergo oxidation of cysteine thiols, was investigated in HEK-293T cells. The work is very relevant, well written and brings information that is very well compiled in well executed graphics. For the paper to be accepted in this journal, I believe that in the introduction section, the authors should better explain the choice of HEK-293T cells. Also, the figures need to be in a resolution more appropriate for the manuscript.
After these minor adjustments, I believe that the manuscript is ready to be accepted for publication.

Author Response

Reviewer 1

 In the present work, the ability of the circadian protein period 2 (PER2) to undergo oxidation of cysteine thiols, was investigated in HEK-293T cells. The work is very relevant, well written and brings information that is very well compiled in well executed graphics. For the paper to be accepted in this journal, I believe that in the introduction section, the authors should better explain the choice of HEK-293T cells. Also, the figures need to be in a resolution more appropriate for the manuscript.

After these minor adjustments, I believe that the manuscript is ready to be accepted for publication.

We agree with the reviewer's comments. HEK-293T cells were selected due to their use as an established model for studying the circadian molecular clock regulation (resetting experiments with dexamethasone, Beta et al., 2022, 10.1186/s13104-021-05871-7; peroxide control of CLOCK cysteine oxidation on CLOCK-BMAL1 heterodimer activity, Pei et al., 10.1038/s41556-019-0420-4), or its outputs (proteasome activity on oxidized proteins, Desvergne et al., 2016, 10.1016/j.freeradbiomed.2016.02.037), (enzymatic function in oxidative stress, Wickramaratne et al., 2022, 10.1021/acs.biochem.2c00072), for giving some examples. We have stated this in the Introduction section (page 2, 3rd paragraph).

In addition, the resolution of all figures was increased as requested.

Reviewer 2 Report

In this study, Baidanoff and coworkers use PEG-switch and Biotin-switch assays to confirm that there are accessible cysteine(s) in PER2 (and CRY2) susceptible to oxidation. Furthermore, the authors characterize the PER2 cysteine oxidation reaction using CysNO and find that this oxidation drives the PER2 oligomerization and subsequently may affect the PER2 protein stability.

Overall, this manuscript is well-written and technically sound. I only have one major

and several minor criticisms.

Major points

1.    In Figure 1A, is this a Westernblot exposure? If yes, the authors should indicate what the primary antibody is used. Also, Page 3, line 131-132, “Increasing gel resolution, as well as using a higher molecular weight PEG, did not generate a greater displacement for PER2 (data not shown)”, data not shown is not acceptable, the authors should include these results as supplementary. Meanwhile, the authors should also discuss this negative result in discuss section. What might be the possible reasons causing this.

Minor points

1.    Page 2, Line 62, change “2.1 Drugs”  to something “Synthesis of S-nitroso-L-cysteine” or “Preparation of S-nitroso-L-cysteine”

2.    Figure 1C, what happens to the lane of CysNO 15 min treatment, the authors should address why this lane is empty.

3.    Figure 1B (a), add the standard molecular weight marker. If possible, try to label where PER2 is (maybe impossible).

4.    My understanding is that the methodology used in result section 3.3 is just treating lysate with CysNO, not biotin-switch assay. If so, the author should write  method details for “Characterization of the cysteine oxidation reaction” in materials and methods section.

5.    I recommend making a new figure (figure 4) to summarize the proposed putative mechanism and importance of this PER2 cysteine oxidation, since a hypothesis is already put forward in the discussion.

6.    In the original images file, there is a mislabel, Figure 3B should be figure 3C.

7.    In the original images file, “figure 3A” is exactly the same as the figure shown in figure 3A, not the original image

Author Response

Reviewer 2

In this study, Baidanoff and coworkers use PEG-switch and Biotin-switch assays to confirm that there are accessible cysteine(s) in PER2 (and CRY2) susceptible to oxidation. Furthermore, the authors characterize the PER2 cysteine oxidation reaction using CysNO and find that this oxidation drives the PER2 oligomerization and subsequently may affect the PER2 protein stability.

Overall, this manuscript is well-written and technically sound. I only have one major

and several minor criticisms.

Major points

  1. In Figure 1A, is this a Westernblot exposure? If yes, the authors should indicate what the primary antibody is used.

We are sorry for the omission. Figure 1A is a Western Blot for PER2. We have now made this clear in the text and in the Figure.

Also, Page 3, line 131-132, “Increasing gel resolution, as well as using a higher molecular weight PEG, did not generate a greater displacement for PER2 (data not shown)”, data not shown is not acceptable, the authors should include these results as supplementary. Meanwhile, the authors should also discuss this negative result in discuss section. What might be the possible reasons causing this.

We agree with the reviewer’s comment. This sentence was added to state that the PEG switch was performed under different conditions, and that the obtained result was the maximum gel displacement possible for PER2. As the reviewer suggest, we have now included a new Supplementary Figure 2 including western blots for PER2 PEG switch assayed under two additional conditions that failed to generate a substantial displacement (as observed for CRY2, Supp. Fig. 1). This is because PER2 is a protein of 130 kDa, which is a large molecular weight for this kind of assay. Increasing PEG molecular weight to 20 kDa is also not recommended. Thus, we decided to include PEG switch for 10 kDa (Figure 1A) showing enough band displacement for deciding to study PER2 as a candidate for cysteine oxidation experiments included in the article. In addition, we have included these statements in the Discussion section in accordance (page 9, 1st paragraph).

Minor points

  1. Page 2, Line 62, change “2.1 Drugs” to something “Synthesis of S-nitroso-L-cysteine” or “Preparation of S-nitroso-L-cysteine”

The section title was changed in accordance with the reviewer’s comment.

  1. Figure 1C, what happens to the lane of CysNO 15 min treatment, the authors should address why this lane is empty.

Treatment with CysNO generated protein S-nitrosation which is unstable and undetectable after 5 min, that is the reason why lane for 15 min is empty, with no signal for S-nitrosated PER2. This was clarified in the Results section (page 4, 2nd paragraph)

  1. Figure 1B (a), add the standard molecular weight marker. If possible, try to label where PER2 is (maybe impossible).

The molecular weight marker was added in the Figure 1B. As the reviewer suggests, we were not able to label PER2 in this western blot, which is revealing total protein S-nitrosations.

  1. My understanding is that the methodology used in result section 3.3 is just treating lysate with CysNO, not biotin-switch assay. If so, the author should write method details for “Characterization of the cysteine oxidation reaction” in the materials and methods section.

A subsection was added in Methods to explain this (page 3, 4th paragraph)

  1. I recommend making a new figure (figure 4) to summarize the proposed putative mechanism and importance of this PER2 cysteine oxidation, since a hypothesis is already put forward in the discussion.

A new Figure 4 was added to summarize the putative mechanism involving PER2 cysteine oxidation in the regulation of the molecular circadian clock.

  1. In the original images file, there is a mislabel, Figure 3B should be figure 3C.

The mislabel was corrected.

  1. In the original images file, “figure 3A” is exactly the same as the figure shown in figure 3A, not the original image

We are sorry for the mistake in the uploaded image. Unfortunately, we do not have the original images of the Figures 1A, 1B, 3A, 3B. This was stated previously to the Journal's Editor.

Reviewer 3 Report

The authors presented data demonstrating that cysteine oxidation is involved in unit dimer- and oligomerization of circadian protein Period2. This in vitro study could provide insights into the mechanisms of posttranslational modification involved in circadian clock operation. I think the presentation of data could be improved if the following is addressed:

1. In Fig. 1A, what is the difference between the two PBS bands on either side of the gel image? It's not clear why the right side PBS gel band has much less intensity when compared to that of the left side PBS gel band.

2. Fig. 3 A gel image: the right panel, what the minus and plus signs on top of the image mean? Do they mean without or with CysNO? Or do they mean yes or no for mercaptoethanol? Please clarify.

3. If the PER2 cysteine modification is modulated by H2O2, do the authors see corresponding ocilation of H2O2 in the HEK-293T cells? An explanation why HEK-293 cells were chosen would also help the readers.

Author Response

Reviewer 3

The authors presented data demonstrating that cysteine oxidation is involved in unit dimer- and oligomerization of circadian protein Period2. This in vitro study could provide insights into the mechanisms of posttranslational modification involved in circadian clock operation. I think the presentation of data could be improved if the following is addressed:

  1. In Fig. 1A, what is the difference between the two PBS bands on either side of the gel image? It's not clear why the right side PBS gel band has much less intensity when compared to that of the left side PBS gel band.

In Fig 1A, there is shown a PEG-switch for PER2 at different H202 concentrations. Both PBS are controls of the reaction. This gel is measuring band displacement, which in both PBS treated samples is the same (no displacement). The reviewer is right to point out that the intensity is different, but this is due to different amount of protein in the sample, which does not have an impact on the PEG-switch assay.

  1. Fig. 3 A gel image: the right panel, what the minus and plus signs on top of the image mean? Do they mean without or with CysNO? Or do they mean yes or no for mercaptoethanol? Please clarify.

We are sorry for the confusion in Figure 3A. The left panel is a western blot for control samples (without 10% 2-mercaptoethanol), and the right one is for samples treated with 10% 2-mercaptoethanol. The minus and plus signs indicate the presence or absence of CysNO in both panels, respectively. The figure was changed for clarity, by adding a legend for 2-mercaptoethanol treatment.

  1. If the PER2 cysteine modification is modulated by H2O2, do the authors see corresponding ocilation of H2O2 in the HEK-293T cells? An explanation why HEK-293 cells were chosen would also help the readers.

We did not study H202 oscillations in HEK-293T cells, and to our knowledge, this was only evidenced in N2a mouse neuroblastoma and human U2OS osteosarcoma (Pei et al., 10.1038/s41556-019-0420-4). We did not assess the circadian rhythm of oxidation of PER2 in HEK cells, but only its ability to be oxidized either by CysNO (likely an S-nitrosation), as well as by H202. To that aim, using HEK cells to overexpress PER2 is a useful tool to obtain substantial amounts of this protein for western blot experiments.

HEK-293T cells were selected due to their use as an established model for studying the circadian molecular clock regulation (resetting experiments with dexamethasone, Beta et al., 2022, 10.1186/s13104-021-05871-7; peroxide control of CLOCK cysteine oxidation on CLOCK-BMAL1 heterodimer activity, Pei et al., 10.1038/s41556-019-0420-4), or its outputs (proteasome activity on oxidized proteins, Desvergne et al., 2016, 10.1016/j.freeradbiomed.2016.02.037), (enzymatic function in oxidative stress, Wickramaratne et al., 2022, 10.1021/acs.biochem.2c00072), for giving some examples. We have stated this in the Introduction section (page 2, 3rd paragraph).